# Seeking Truth and Beauty in Flavor Physics with Machine Learning

**Konstantin T. Matchev**
Institute for Fundamental Theory
Department of Physics
University of Florida
Gainesville, FL 32611
matchev@ufl.edu

**Katia Matcheva**
Institute for Fundamental Theory
Department of Physics
University of Florida
Gainesville, FL 32611
matcheva@ufl.edu

**Pierre Ramond**
Institute for Fundamental Theory
Department of Physics
University of Florida
Gainesville, FL 32611
ramond@phys.ufl.edu

**Sarunas Verner**\*
Institute for Fundamental Theory
Department of Physics
University of Florida
Gainesville, FL 32611
verner.s@ufl.edu

## Abstract

The discovery process of building new theoretical physics models involves the dual aspect of both fitting to the existing experimental data and satisfying abstract theorists' criteria like beauty, naturalness, etc. We design loss functions for performing both of those tasks with machine learning techniques. We use the Yukawa quark sector as a toy example to demonstrate that the optimization of these loss functions results in true and beautiful models.

> **Murdock:** Rambo, you can feel totally safe because we have the most advanced weapons in the world available to us.
> **Rambo:** I've always believed that the mind is the best weapon.
> **Murdock:** Times change.
> **Rambo:** For some people.
>
> *Rambo: First Blood Part II (1985)*

## 1 Introduction

Ever since ancient times, great technological progress in human society has been accompanied by equally impressive intellectual leaps by the great minds of the previous generations. With the recent boom in artificial intelligence, the machines are beginning to challenge humans even at tasks typically reserved for "deep thinkers". Nowhere is this dichotomy more evident than in the field of theoretical physics, as theoretical physicists like Isaac Newton, Albert Einstein, etc., regularly top the lists of smartest people of all time [1].

The task of a theoretical physicist is to develop a theory model describing a set of natural physics phenomena. There are two aspects of this process:

---

\*Corresponding and first author; remaining authors have equal contributions.

37th Conference on Neural Information Processing Systems (NeurIPS 2023).

- *Truth.* Above all, the model has to be truthful, in the sense that it can correctly account for the existing set of measurements of a number of experimental observables $\{\mathcal{O}_\alpha\}$, $\alpha = 1, 2, \ldots, N_\mathcal{O}$. This is accomplished by tuning the model parameters $\{\mathcal{P}_i\}$, $i = 1, 2, \ldots, N_\mathcal{P}$, until the model predictions fit the data. This adjustment can typically be done rather easily, since in most models the number of tunable model parameters exceeds the number of available measurements, i.e., $N_\mathcal{P} > N_\mathcal{O}$.

- *Beauty.* After this fitting procedure, we are typically still left with a number of model parameters (namely, $N_\mathcal{P} - N_\mathcal{O}$) which cannot be determined from data. Instead, they can be chosen to make the model more "beautiful". However, this is the point where typically one encounters a number of different opinions (after all, beauty is in the eye of the beholder). Since beauty is an inherently subjective concept, different theorists, guided by their own theoretical prejudices, could easily disagree to what extent a given theory model is "beautiful".

From a machine learning standpoint, there is nothing mysterious about "beauty", as long as it can be quantified, i.e., there exists an agreed-upon beforehand, community-wide quantitative measure indicating the "beauty" of a model. In the past such measures have been introduced to quantify the fine-tuning in new physics models like low-energy supersymmetry [2–6]. Once a quantitative measure of the model's beauty is adopted, model building becomes a simple optimization problem amenable to machine learning approaches.

In this paper, we focus on the flavor sector, which is arguably the "ugliest" part of the Standard Model. New physics models can therefore offer many opportunities for improvement on the "beauty" scale. We consider several possible choices for quantitative measures of the beauty of the model. In each case, we define corresponding loss functions whose minimization by construction yields "the most truthful and beautiful" model.

Our approach should be viewed as part of a much broader program of trying to learn the laws of nature with a machine, eliminating any human intervention whatsoever [7–17]. For example, it has been demonstrated that the machine can re-derive the known classical physics laws from data [18–21]. Symbolic learning was recently successfully applied to problems in a wide range of physics areas, e.g. in astrophysics [22, 19, 23], in astronomy for the study of orbital dynamics [24, 25] and exoplanet transmission spectroscopy [26], in collider physics [27–31], in materials science [32], and in behavioral science [33]. Our approach is slightly less ambitious than those studies, since we already assume the mathematical framework for the description of the phenomena, and instead focus only on the determination of the "best" model parameters which, within that mathematical framework, might have been chosen by nature.

The paper is organized as follows. In Section 2, we introduce our notation and provide the minimal particle physics background needed to understand the results in the sections to follow. Then in Section 3 we consider two examples of "beautiful" quark textures. First, in Section 3.1 we consider beauty to mean uniformity, i.e., the elements in the Yukawa matrices have the same magnitude. Then in Section 3.2 we take beauty to mean sparsity, i.e., the Yukawa matrices have a large number of vanishing elements. Section 4 contains our summary and conclusions.

## 2   Standard Model Parameters

For the most part, we use the notation in the standard textbook [34]. The Lagrangian of the quark mass sector is

$$\mathcal{L}_{\text{quarks}} = -Y_{ij}^d \bar{Q}^i H d_R^j - Y_{ij}^u \bar{Q}^i \widetilde{H} u_R^j + \text{h.c.} \tag{1}$$

Here $Q^i$, $i = 1, 2, 3$ are the three families of $SU(2)_L$ quark doublets,

$$Q^i = \begin{pmatrix} u_L^i \\ d_L^i \end{pmatrix}, \tag{2}$$

$H$ is the Higgs field and $\tilde{H}$ is its conjugate given by

$$\tilde{H} \equiv i\sigma_2 H^*, \tag{3}$$

where $\sigma_2$ is the second Pauli matrix and $*$ denotes complex conjugation. Explicitly,

$$H = \begin{pmatrix} H^+ \\ H^0 \end{pmatrix}, \quad \tilde{H} = \begin{pmatrix} H^{0*} \\ -H^- \end{pmatrix}. \tag{4}$$

The fields $u_R^i$, and $d_R^i$ are the $SU(2)_L$ quark singlets

$$u_R^i = \{u_R, c_R, t_R\}, \qquad d_R^i = \{d_R, s_R, b_R\}. \tag{5}$$

From here on throughout we include all three generations of quarks. In general, the Yukawa matrices $Y_{ij}^u$ and $Y_{ij}^d$ are arbitrary complex matrices and do not have to obey hermiticity or any other such mathematical properties. After spontaneous symmetry breaking, the neutral component $H^0$ of the Higgs field obtains a vacuum expectation value $v/\sqrt{2}$ and the quark mass Lagrangian (1) becomes

$$\mathcal{L}_{\text{quarks}} = -\frac{v}{\sqrt{2}} \left[ \bar{d}_L Y^d d_R + \bar{u}_L Y^u u_R \right] + \text{h.c.} \tag{6}$$

In this so-called interaction eigenstate basis, the quark mass matrices

$$(M_u)_{ij} \equiv \frac{v}{\sqrt{2}} Y_{ij}^u, \qquad (M_d)_{ij} \equiv \frac{v}{\sqrt{2}} Y_{ij}^d \tag{7}$$

are not diagonal in flavor space, but the interactions of the up and down types quarks to the $W^\pm$-bosons are.

The mass matrices (7) can be diagonalized by performing a change of basis

$$u_L' = U_u u_L, \qquad d_L' = U_d d_L, \tag{8a}$$
$$u_R' = K_u u_R, \qquad d_R' = K_d d_R. \tag{8b}$$

Here the primed quark fields are the mass eigenstates, while $U_u$, $U_d$, $K_u$ and $K_d$ are unitary rotation matrices. The mass matrices in this new basis are diagonal and are given by

$$M_u' = U_u M_u K_u^\dagger, \qquad M_d' = U_d M_d K_d^\dagger. \tag{9}$$

The matrices $U_u$ and $K_u$ can be found by diagonalizing the Hermitian matrices $M_u M_u^\dagger$ and $M_u^\dagger M_u$, respectively. Since those matrices are Hermitian, they have three mass (more precisely, mass-squared) eigenvalues, which correspond to the three up-type quark masses $m_i^u \equiv \{m_u, m_c, m_t\}$ for the up quark, charm quark and top quark, respectively.

Similarly, the matrices $U_d$ and $K_d$ can be found by correspondingly diagonalizing $M_d M_d^\dagger$ and $M_d^\dagger M_d$, which yields the masses $m_i^d \equiv \{m_d, m_s, m_b\}$ for the down, strange and bottom quark, respectively.

With this notation, the quark mass Lagrangian in the mass eigenstate basis becomes diagonal in flavor space:

$$\mathcal{L}_{\text{quarks}} = -m_i^u \delta_{ij} \bar{u}_L'^i u_R'^j - m_i^d \delta_{ij} \bar{d}_L'^i d_R'^j + \text{h.c.}, \tag{10}$$

but the quark mixing effects reappear in the interaction vertices involving charged $W$ bosons. Those are conventionally parameterized with the Cabibbo-Kobayashi-Maskawa (CKM) matrix:

$$V_{\text{CKM}} \equiv U_u^\dagger U_d = \begin{pmatrix} V_{11} & V_{12} & V_{13} \\ V_{21} & V_{22} & V_{23} \\ V_{31} & V_{32} & V_{33} \end{pmatrix} = \begin{pmatrix} V_{ud} & V_{us} & V_{ub} \\ V_{cd} & V_{cs} & V_{cb} \\ V_{td} & V_{ts} & V_{tb} \end{pmatrix}. \tag{11}$$

The CKM matrix is a $3 \times 3$ complex unitary matrix with a total of 9 degrees of freedom. By using the $U(1)$ symmetry of the 6 mass terms in the Lagrangian (10), we can eliminate 5 degrees of freedom[2], leaving us with only 4 degrees of freedom which are often parametrized in the following way

$$V_{\text{CKM}} = \begin{pmatrix} 1 & 0 & 0 \\ 0 & \cos\theta_{23} & \sin\theta_{23} \\ 0 & -\sin\theta_{23} & \cos\theta_{23} \end{pmatrix} \begin{pmatrix} \cos\theta_{13} & 0 & \sin\theta_{13} e^{i\delta} \\ 0 & 1 & 0 \\ -\sin\theta_{13} e^{i\delta} & 0 & \cos\theta_{13} \end{pmatrix} \begin{pmatrix} \cos\theta_{12} & \sin\theta_{12} & 0 \\ -\sin\theta_{12} & \cos\theta_{12} & 0 \\ 0 & 0 & 1 \end{pmatrix}$$

$$= \begin{pmatrix} c_{12}c_{13} & s_{12}c_{13} & s_{13}e^{-i\delta} \\ -s_{12}c_{23} - c_{12}s_{23}s_{13}e^{i\delta} & c_{12}c_{23} - s_{12}s_{23}s_{13}e^{i\delta} & s_{23}c_{13} \\ s_{12}s_{23} - c_{12}c_{23}s_{13}e^{i\delta} & -c_{12}s_{23} - s_{12}c_{23}s_{13}e^{i\delta} & c_{23}c_{13} \end{pmatrix}, \tag{12}$$

where $s_{ij} \equiv \sin\theta_{ij}$ and $c_{ij} \equiv \cos\theta_{ij}$, and $\delta$ is the CP-violating phase. Assuming unitarity, the CKM matrix parameters are given by [35]

$$\sin\theta_{12} = 0.22500 \pm 0.00067, \qquad \sin\theta_{13} = 0.00369 \pm 0.00011,$$
$$\sin\theta_{23} = 0.04182^{+0.00085}_{-0.00074}, \qquad \delta = 1.144 \pm 0.027.$$

---

[2]We cannot eliminate all 6 degrees of freedom because the phases of the $U(1)$ rotations must be different. Otherwise, the CKM matrix remains unchanged.

Table 1: Quark masses (with uncertainties) evaluated at the top quark mass scale.

| $m_u$ (MeV) | $m_d$ (MeV) | $m_c$ (GeV) | $m_s$ (MeV) | $m_b$ (GeV) | $m_t$ (GeV) |
|---|---|---|---|---|---|
| $1.22^{+0.28}_{-0.15}$ | $2.76^{+0.28}_{-0.10}$ | $0.59^{+0.01}_{-0.01}$ | $52^{+4.79}_{-1.89}$ | $2.75^{+0.02}_{-0.01}$ | $162.9^{+0.28}_{-0.28}$ |

The fit results for the magnitudes of all nine CKM elements are

$$|V_{\mathrm{CKM,exp}}| = \begin{pmatrix} 0.97435 \pm 0.00016 & 0.22500 \pm 0.00067 & 0.00369 \pm 0.00011 \\ 0.22486 \pm 0.00067 & 0.97349 \pm 0.00016 & 0.04182^{+0.00085}_{-0.000074} \\ 0.00857^{+0.00020}_{-0.00018} & 0.0410^{+0.0003}_{-0.00072} & 0.999118^{+0.0000031}_{-0.000036} \end{pmatrix} . \tag{13}$$

In addition to (13), the other inputs in our analysis will be the running quark masses evaluated at some reference energy scale. We choose the top quark mass scale [36] (other choices of a reference scale are possible as well, for example the $Z$ mass scale [37]) and summarize the corresponding values with their experimental uncertainties in Table 1.

## 3  Quark Sector Textures

In this section, we build our loss functions and demonstrate their utility with two examples. The inputs to the original Lagrangian (1) are the Yukawa matrices $Y^u$ and $Y^d$, or equivalently, the corresponding mass matrices $M_u$ and $M_d$, which have a total of 36 degrees of freedom. Out of those, 9+6=15 are fixed by the experimental inputs in Eq. (13) and Table 1. So in principle, we could set this up as an optimization problem in 36 dimensions, subject to 15 constraints. However, to accelerate the optimization, we choose a parametrization for $M_u$ and $M_d$ which manifestly solves the quark mass constraints, namely the inverse relations to (9):

$$M_u = U_u^\dagger M_u' K_u, \qquad M_d = U_d^\dagger M_d' K_d. \tag{14}$$

By taking the diagonal entries in the matrices $M_u'$ and $M_d'$ to have magnitudes equal to the respective quark masses, the quark mass constraints are automatically satisfied. This leaves us with only 18-3=15 degrees of freedom in each matrix $M_u$ and $M_d$ (to exactly match the count of degrees of freedom between the LHS and the RHS of the equations in (14), we fix two degrees of freedom in each rotation matrix $U_u$, $U_d$, $K_u$ and $K_d$ by hand). In other words, we have equivalently reformulated the problem as optimization in 30 dimensional space subject only to the 9 constraints (13). To ensure that those are satisfied, we choose the following loss function

$$L_{\mathrm{CKM}} = \sum_{ij} \left( |V_{\mathrm{CKM}}|_{ij} - |V_{\mathrm{CKM,exp}}|_{ij} \right)^2 . \tag{15}$$

Each of the two examples below will be illustrated with a number of pseudo-experiments. In each pseudo-experiment, we shall start not with the central values for the inputs in Eq. (13) and Table 1, but with a set of experimental inputs sampled from split normal distributions with the left (right) standard deviation given by the lower (upper) experimental uncertainty on the corresponding experimental quantity. In what follows, we choose to quote our results in terms of the mass matrices $M_u$ and $M_d$ rather than the dimensionless Yukawas $Y^u$ and $Y^d$.

### 3.1  Uniform Texture

The origin of the Yukawa matrices $Y^u$ and $Y^d$ is one of the major unresolved puzzles in the Standard Model (SM). This so-called flavor problem is an active area of theoretical research for the past 50 years. Many proposed solutions for these "Yukawa textures" exist on the market, and they typically involve new symmetries, new particles and new interactions. Ultimately, the fate of these new physics models will be decided by experiment, by either finding or ruling out those additional structures. Here we consider a bottom-up approach within the SM as an effective theory, where the only experimental measurements available to us are those of Eq. (13) and Table 1. In that case, our only guiding principle in choosing one model over the other is whether the resulting Yukawa sector is "beautiful" or not.

As a warm-up exercise, let us declare that a "beautiful" flavor model is one which predicts uniformity, i.e., all elements in a given Yukawa matrix have (roughly) equal magnitudes, e.g.

$$|Y^u_{ij}| \simeq |Y^u_{kl}|, \qquad \forall i,j,k,l . \tag{16}$$

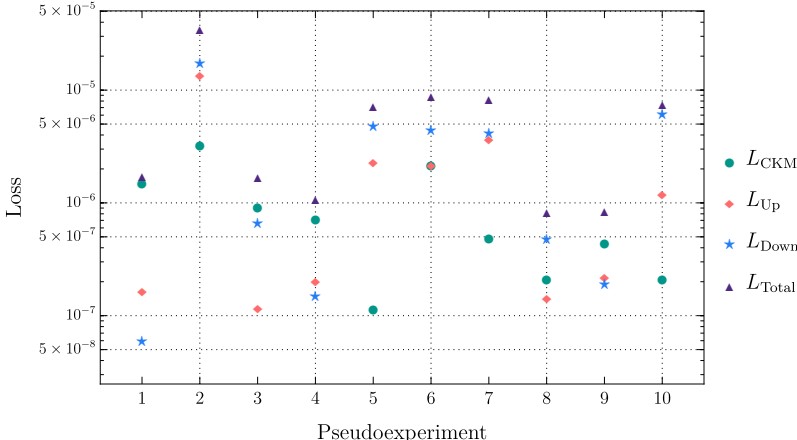

Figure 1: The values for the trained total loss (20) and the breakdown of the three individual contributions (15), (17) and (19), in 10 representative pseudo-experiments for the uniform texture exercise considered in Section 3.1.

This condition can be enforced by introducing the following loss function

$$L_{\text{const,up}} = \sum_{i,j,k,l} \left( |Y_{ij}^u| - |Y_{kl}^u| \right)^2 . \tag{17}$$

Similarly, uniformity for the down-type Yukawa matrix implies

$$|Y_{ij}^d| \simeq |Y_{kl}^d| , \qquad \forall i,j,k,l , \tag{18}$$

and the corresponding loss function is

$$L_{\text{const,down}} = \sum_{i,j,k,l} \left( |Y_{ij}^d| - |Y_{kl}^d| \right)^2 . \tag{19}$$

Therefore, the full loss function for the uniform Yukawa textures is given by

$$L_{\text{uniform}} = L_{\text{CKM}} + \frac{1}{m_t} L_{\text{const,up}} + \frac{1}{m_b} L_{\text{const,down}} , \tag{20}$$

where we normalized the loss functions $L_{\text{const,up}}$ and $L_{\text{const,down}}$ with respect to the heaviest quarks to ensure that all three contributions in the loss function have similar weights. We perform 10 pseudoexperiments and minimize the full loss function (20). The results are shown in Fig. (1), where in addition to the trained values for the total loss we also list the individual contributions (15), (17) and (19). We observe that in each pseudo-experiment we are able to achieve very small values for the loss, indicating viable Yukawa textures.

For illustration, we quote one particular result from the 10 pseudoexperiments (the others are very similar). For the up-type quark mass matrix, we find

$$M_u = \begin{pmatrix} 37.4241 + 39.2292i & -18.2387 + 51.0572i & -52.4459 - 13.7506i \\ 37.9035 + 38.9229i & -17.2426 + 51.3899i & -52.6902 - 12.9535i \\ 42.4129 + 33.5949i & -11.5970 + 52.9758i & -53.7599 - 6.6929i \end{pmatrix} , \tag{21}$$

and

$$|M_u| = \begin{pmatrix} 54.3452 & 54.3444 & 54.3459 \\ 54.4079 & 54.2766 & 54.3340 \\ 54.2823 & 54.4144 & 54.3556 \end{pmatrix} . \tag{22}$$

Similarly, for the down-type mass matrix $M_d$, we obtain

$$M_d = \begin{pmatrix} 0.3461 - 0.8576i & 0.5714 + 0.7318i & 0.2355 + 0.8896i \\ 0.2747 - 0.8801i & 0.6929 + 0.6161i & 0.3569 + 0.8483i \\ 0.2450 + 0.8938i & -0.9014 - 0.2191i & -0.7375 - 0.5554i \end{pmatrix} , \tag{23}$$

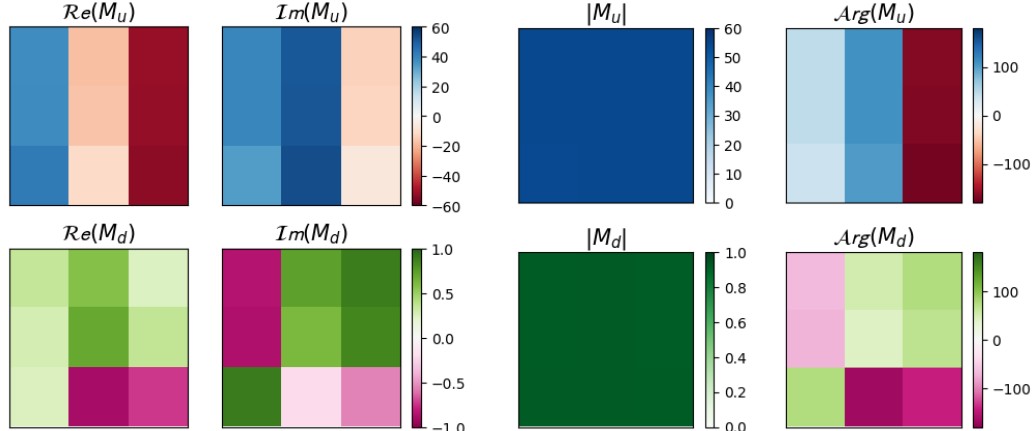

Figure 2: The learned mass matrices $M_u$ (top panels) and $M_d$ (bottom panels) for the uniform Yukawa texture example considered in Sec. 3.1. Each panel represents a learned matrix, where the values of the individual elements of the matrix are indicated by the color bar.

and

$$|M_d| = \begin{pmatrix} 0.9238 & 0.9260 & 0.9210 \\ 0.9238 & 0.9251 & 0.9218 \\ 0.9233 & 0.9251 & 0.9226 \end{pmatrix}. \tag{24}$$

We pictorially illustrate the results from this pseudo-experiment in Fig. 2, where the top panels correspond to (21) and the bottom panels correspond to (23). In each row, the first two panels on the left show the real and imaginary part of the respective matrix element, while the last two panels show its magnitude and phase. We see that in each mass matrix, the magnitudes of the different elements are equal to a very good approximation, which was the criterion for "beauty" in this example.

## 3.2 Zero Textures

For our second example, we shall consider the so-called zero textures [38], where the "beauty" of the model is measured in terms of sparsity. The idea is to have as many vanishing elements in the Yukawa matrices as possible. In our study, we shall take the number of such vanishing elements $N$ as a hyperparameter whose value can be varied, see Table 2.

Once we fix the value of $N$, we still have the freedom to choose exactly which $N$ elements in the matrix are zero. In general, the number of such patterns is given in the second row of Table 2, but some of them are unacceptable because they automatically result in at least one zero mass eigenvalue. The number of remaining, potentially acceptable, patterns is listed in the last row of the table.

For the purposes of this study, we focus on a few representative examples shown in Fig. 3, where circles indicate the locations of the matrix elements which are required to vanish. Let $\mathcal{S}$ represent the set of zero locations in a given pattern, e.g., $\mathcal{S} = \{11, 22, 13\}$ for the first $N = 3$ pattern in Fig. 3. The corresponding loss functions are then given by

$$L_{\text{zeros,up}} = \sum_{ij \in \mathcal{S}} |Y_{ij}^u|^2, \tag{25}$$

Table 2: For a given number $N$ of vanishing elements in the mass matrix (top row), the total number of patterns (middle row) and the number of potentially acceptable patterns (bottom row).

| N | 1 | 2 | 3 | 4 | 5 | 6 | 7 | 8 |
|---|---|---|---|---|---|---|---|---|
| All patterns | 9 | 36 | 84 | 126 | 126 | 84 | 36 | 9 |
| Acceptable | 9 | 36 | 78 | 81 | 36 | 6 | 0 | 0 |

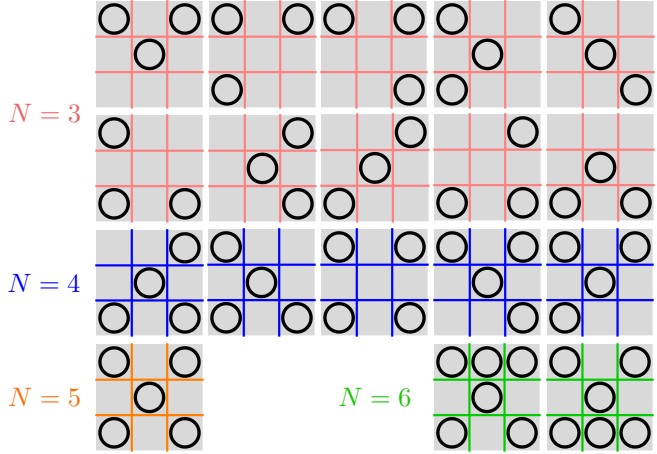

Figure 3: The zero texture patterns considered in the example of Sec. 3.2.

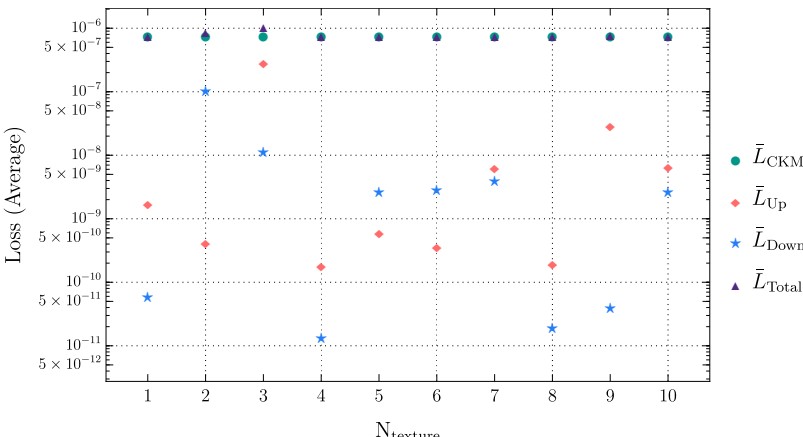

Figure 4: The same as Fig. 1, but for the 10 $N = 3$ zero texture patterns in Fig. 3, averaged over 10 different pseudo-experiments.

and

$$L_{\text{zeros,down}} = \sum_{ij \in \mathcal{S}} |Y^d_{ij}|^2 . \tag{26}$$

We then minimize the full loss function

$$L = L_{\text{CKM}} + \frac{1}{m_t} L_{\text{zeros,up}} + \frac{1}{m_b} L_{\text{zeros,down}} . \tag{27}$$

Once again, we find that viable patterns result in low loss values. In complete analogy to Fig. 1, in Fig. 4 we show the values of the trained loss and its components, for the 10 $N = 3$ zero texture patterns in Fig. 3, averaged over 10 different pseudo-experiments. As expected, all of the $N = 3$ zero texture patterns are possible. The result from one pseudo-experiment for the very first pattern in Fig. 3 is given by

$$M_u = \begin{pmatrix} 0.0001 - 0.0008i & -0.2209 - 0.0169i & 0.0000 + 0.0000i \\ -0.8953 - 0.1084i & 0.0000 + 0.0000i & -9.2195 + 1.1346i \\ 15.4244 + 3.1818i & 9.4056 + 0.7511i & 161.3240 - 6.3096i \end{pmatrix} \tag{28}$$

and

$$M_d = \begin{pmatrix} -0.0000 + 0.0000i & -0.0167 + 0.0105i & 0.0000 + 0.0000i \\ -0.0068 - 0.0008i & -0.0000 + 0.0000i & -0.0724 + 0.0027i \\ 0.5029 + 0.0106i & 2.1104 - 0.7527i & 1.5131 - 0.1968i \end{pmatrix} . \tag{29}$$

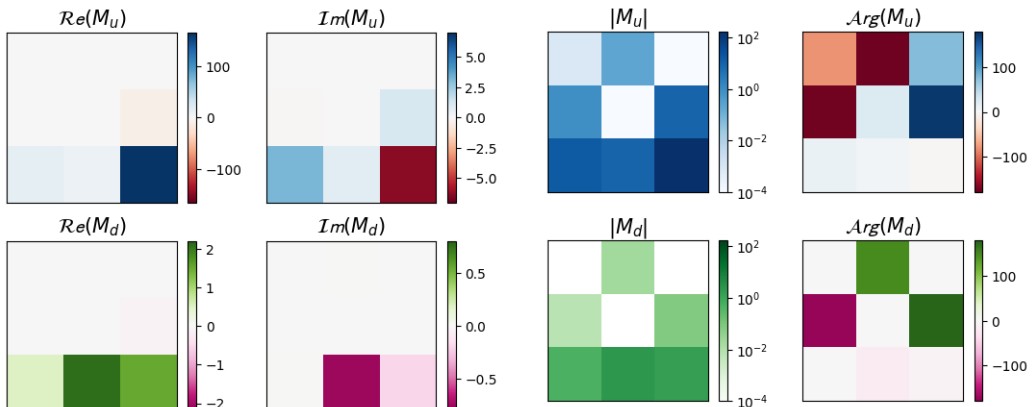

Figure 5: The same as Fig. 2, but for the three zero texture result in Eqs. (28) and (29).

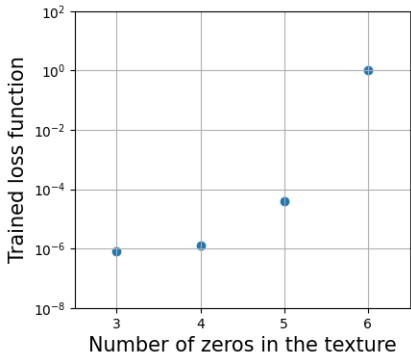

Figure 6: Average value of the trained loss as a function of the number of zeros in the texture. Note the 2-3 orders of magnitude difference between the case of $N = 6$ and the rest.

This result is pictorially illustrated in Fig. 5 and confirms that the entries in positions 11, 22 and 13 are very small (see the third panels in each row).

The results for higher values of $N$ are summarized in Fig. 6, where we plot the values of the trained loss averaged over both the number of pseudo-experiments (in this case 10) and the different patterns in Fig. 3 corresponding to that particular value of $N$. Judging by the values of the loss, we conclude that zero textures with $N = 3, 4, 5$ are possible, while the two patterns with $N = 6$ are ruled out.

## 4 Summary and Conclusions

In the realm of scientific inquiry, the process of developing novel theoretical physics models entails meeting the objective demands of the existing experimental data, as well as the subjective criteria like beauty and naturalness set forth by the theoretical physics community. To achieve both of these objectives, we employ machine learning techniques with suitably designed loss functions addressing the perceived deficiencies in the Yukawa sector of the Standard Model. With a couple of toy examples, we showed that this approach yields models that are not only consistent with the experimental data, but also possess the desired aesthetic elegance as defined by a quantitative benchmark.

## Acknowledgments and Disclosure of Funding

The work of KTM, PR and SV is supported in part by an U.S. Department of Energy award number DE-SC0022148.

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
