# OpenReview forum: "Seeking Truth and Beauty in Flavor Physics with Machine Learning"
_NeurIPS.cc/2023/Workshop/AI4Science — NeurIPS2023-AI4Science Poster_

### Official Review · Reviewer_E6nr · 2023-10-24
**Review of "Seeking Truth and Beauty in Flavor Physics with Machine Learning"**

**Rating:** 6
**Confidence:** 3

**Review:**

This work looks at employing machine learning techniques (with suitably chosen loss functions) to address the perceived deficiencies in the Yukawa sector of the Standard Model.

They demonstrate that their approach gives rise to models which are consistent with the experimental data and are sufficiently elegant, as measured by a quantitative benchmark. Specifically, they focus on the flavor sector and look at two examples of "beautiful" quark textures under two distinct definitions of "beauty" (uniformity and sparsity, respectively).

They emphasize that this approach should be interpreted as being part of a larger (general) effort of the automated learning of physical laws by a machine.

Specifically, they assume that there is an existing mathematical framework that accurately describes the system of interest, and focus exclusively on determining the "best" model parameters without sacrificing correctness.

Overall, this work proposes an interesting approach that looks at quantifying and improving the "beauty" of physical models. This is done by considering different definitions of beauty (e.g. sparsity and uniformity), and running an optimization procedure to then improve it.

---

### Meta-Review · Area_Chair_DXBA · 2023-10-27

**Recommendation:** Accept (Oral)
**Confidence:** 4

**Metareview:**

I found this paper is pleasant to read and very original.

In this work, the authors propose an interesting approach that looks at quantifying and improving the "beauty", intended as sparsity and uniformity, of physical models. Their results, focusing on the Yukawa Model, agree with experimental data.

Given the originality of the paper and the engaging writing, I recommend acceptance and I'd be curious to see it presented as an oral.